# Contrastive Learning with Quantum Projection Heads and Kernels

## Abstract

Self-supervised contrastive learning is sensitive to architectural choices and to how similarity is defined. Motivated by claims that quantum circuits can induce useful non-classical geometries, we present a systematic empirical analysis of two natural drop-in quantum components for the projection/similarity stage: (i) variational quantum circuits (VQCs) as projection heads and (ii) fixed quantum feature maps whose state fidelities act as similarity measures ("quantum kernels"). Within a controlled SimCLR pipeline on STL-10 (ResNet18 encoder) using mainstream *analytic* simulators, we report three findings. First, under realistic resource constraints (low qubit count, shallow depth), a tuned classical MLP head consistently matches or outperforms VQC heads. Second, fidelity-based quantum kernels largely mirror cosine similarity without a clear uplift. Third, increasing circuit size rapidly incurs prohibitive latency, exposing scaling bottlenecks that restrict current explorability. These results constitute a useful null baseline for hybrid quantum-classical contrastive learning and point to concrete directions: batching-friendly simulators for higher throughput, lower-variance/better-conditioned feature maps to avoid similarity collapse, and modest, low-latency hardware as a realistic near-term testbed. We release anonymized code and consolidated hyperparameters to facilitate replication and future extensions.

## 1 Introduction

Self-supervised contrastive learning has become a standard recipe for building strong visual representations without labels, with simple architectural choices (e.g., a ResNet encoder and a small MLP projection head) delivering competitive performance across tasks (Chen et al., 2020; He et al., 2020; Grill et al., 2020). A line of work—SimCLR, MoCo, and BYOL—established that strong augmentations plus a lightweight projection head, optimized with either a large-batch contrastive objective (SimCLR), a momentum encoder with a queue to supply abundant negatives (MoCo), or even a negative-free bootstrap objective with a target network and predictor (BYOL), can yield representations that rival or surpass supervised pretraining on downstream tasks (Chen et al., 2020; He et al., 2020; Grill et al., 2020). In parallel, enthusiasm for quantum machine learning (QML) has grown rapidly. Variational quantum circuits (VQCs) and quantum feature maps promise non-classical geometries, entanglement-enabled correlations, and kernel families that could be hard to emulate classically (Schuld & Killoran, 2019; Havlíček et al., 2019; Schuld et al., 2020; Cerezo et al., 2021). In particular, Schuld & Killoran (2019) formalized quantum feature maps and kernels (the "quantum kernel trick"), Havlíček et al. (2019) demonstrated supervised learning with quantum-enhanced feature spaces via instantaneous quantum polynomial time (IQP) style circuits and argued for potential classical intractability, Schuld et al. (2020) introduced circuit-centric quantum classifiers based on parameterized rotations and entanglement trained by gradient methods, and Cerezo et al. (2021) surveyed variational quantum algorithms, highlighting expressivity alongside optimization challenges such as barren plateaus. This "quantum optimism" naturally raises the question: can we swap a classical projection head with a quantum component, or compute similarities with quantum-induced fidelities, and obtain a tangible benefit in representation learning?

This paper takes a careful, empirical view. We re-implement a modern contrastive pipeline (ResNet18 backbone (He et al., 2015), InfoNCE loss (van den Oord et al., 2019), STL-10 (Coates et al., 2011)) and replace the projection head with (i) a VQC head and (ii) a fixed quantum feature map used as a fidelity-based similarity inside InfoNCE. We benchmark these against a tuned

classical MLP head of similar parameter count, as well as a minimal "bottleneck linear" control that removes nonlinearity in the head. Our implementations use mainstream toolchains (PyTorch + PennyLane/Qiskit) and reproduce common training protocols and augmentations. The goal is not to claim quantum advantage, but to establish a clean, auditable baseline for where quantum components help, or fail to help, today.

Our contribution is summarized as follows:

- **A transparent, end-to-end contrastive baseline** on STL-10 (ResNet18 + InfoNCE) that evaluates variational quantum circuit (VQC) projection heads and fidelity-based quantum kernels *apples-to-apples* against classical heads under controlled settings.

- **A clear empirical null result under realistic simulator constraints:** VQC heads do not outperform a tuned MLP head, fidelity-based quantum kernels largely mirror cosine similarity, and scaling circuit size rapidly incurs prohibitive latency.

- **A standardized, reproducible baseline with resource-scaling analysis:** anonymized code release with fixed seeds and consolidated hyperparameters to ensure exact replication and facilitate future quantum–classical contrastive studies.

Taken together, our findings suggest that quantum components are not a panacea for contrastive learning today, while outlining concrete simulator and algorithmic improvements that could make hybrid approaches competitive as tooling advances.

## 2 METHODS

### 2.1 EXPERIMENT 0: CLASSICAL BASELINE

We adopt a ResNet-18 encoder (He et al., 2015; Deng et al., 2009) from `torchvision`. In the pretrained-frozen regime, the encoder is initialized with ImageNet-pretrained weights and frozen during training so that only the projection head is updated. In the from-scratch regime, the encoder is initialized randomly and optimized jointly with the projection head using the contrastive loss. The baseline projection head is a 2-layer MLP of the form:

$$\text{MLP}(h) = W_2 \, \sigma(W_1 h + b_1) + b_2, \tag{1}$$

with hidden dimension 512, output dimension 128, and ReLU nonlinearity $\sigma$. We train using the InfoNCE loss:

$$\mathcal{L}_{\text{InfoNCE}} = -\sum_{i=1}^{B} \log \frac{\exp\big(\text{sim}(z_i, z_i^+)/\tau\big)}{\sum_{j=1}^{2B} \mathbf{1}_{[j \neq i]} \exp(\text{sim}(z_i, z_j)/\tau)}, \tag{2}$$

where sim denotes cosine similarity, $\tau$ is the temperature, and $B$ is the batch size. We adopt the SimCLR augmentation strategy (random resized crop, color jitter, grayscale, Gaussian blur, horizontal flip) and train with Adam optimizer (Kingma & Ba, 2017), learning rate $3 \times 10^{-4}$, weight decay $10^{-6}$, batch size 256, and $\tau = 0.1$.

### 2.2 EXPERIMENT 1: VARIATIONAL QUANTUM CIRCUIT (VQC) PROJECTION HEAD

Variational quantum circuits (VQCs) are parameterized quantum circuits built from single–qubit rotations and entangling gates, with parameters optimized by gradient-based methods. They have been explored as expressive function approximators in quantum machine learning (Schuld et al., 2020; Cerezo et al., 2021). In contrastive learning, a VQC can be used as a projection head in place of a classical MLP, potentially inducing different inductive biases via non-classical geometry and entanglement. Our goal is to assess when, and under which readout choices, a VQC head is competitive or beneficial.

Given encoder features $h \in \mathbb{R}^d$, a linear map produces $n_{\text{qubits}}$ angles that are angle–encoded onto $n_{\text{qubits}}$ via single–qubit rotations (e.g., $R_Y$). The circuit applies $n_{\text{layers}}$ blocks, each consisting of trainable single–qubit rotations followed by ring entanglement. An initial readout measured only Pauli–$Z$ expectations, yielding $z_q \in \mathbb{R}^{n_{\text{qubits}}}$. Gradient inspection revealed near–zero updates through the angle–preprocessing layer and very small VQC parameter gradients ($10^{-4}$–$10^{-3}$), with accuracy

largely insensitive to qubit count or depth, indicating a dead–wire failure mode. To increase the effective feature capacity of the readout, we expand the measurement set to Pauli–$X$, Pauli–$Y$, and Pauli–$Z$ expectations per qubit, producing

$$z_q \;=\; \big(\langle X_1 \rangle, \langle Y_1 \rangle, \langle Z_1 \rangle, \dots, \langle X_{n_q} \rangle, \langle Y_{n_q} \rangle, \langle Z_{n_q} \rangle\big) \in \mathbb{R}^{3n_{\text{qubits}}}.$$

A linear lift maps this multi–observable vector to the baseline projection dimension,

$$z \;=\; W_{\text{lift}}\, z_q + b_{\text{lift}}, \qquad \|z\|_2 = 1,$$

after which cosine similarity is used in the InfoNCE objective exactly as in the classical baseline.

### 2.3 EXPERIMENT 2: QUANTUM KERNEL FEATURE MAPPING

We replace the projection head with a fixed quantum feature map $U_\phi$ that embeds encoder features $h \in \mathbb{R}^d$ into an $n_q$-qubit state $|\psi(h)\rangle = U_\phi(a(h)) |0\rangle^{\otimes n_q}$, where $a(h)$ linearly maps $h$ to per-qubit rotation angles. The map uses angle encoding with depth $L$ and ring entanglement; layers consist of single-qubit rotations followed by CZ gates. Similarity between samples $i, j$ is the state fidelity (see (Nielsen & Chuang, 2010))

$$K_{ij} = \big| \langle \psi(h_i) \mid \psi(h_j) \rangle \big|^2,$$

computed analytically on a differentiable simulator from the $2B$ states in each minibatch. We integrate $K$ into contrastive learning by replacing cosine similarity with $K/\tau$ inside InfoNCE (standard NT-Xent layout over $2B$ items, diagonal masked), keeping the encoder, data pipeline, optimizer, and schedules identical to the classical baseline. Unless otherwise stated, $n_q \in \{4, 8\}$, $L \in \{1, 2, 3\}$, angles are squashed via $\tanh(\cdot) \cdot \pi$, and temperatures $\tau \in \{0.5, 0.3, 0.2\}$ are swept. All hyperparameters and exact versions are consolidated in Appendix A.

## 3 RESULTS

### 3.1 EXPERIMENT 0: CLASSICAL BASELINE

We evaluate the classical baseline on STL-10 in two regimes: pretrained–frozen (encoder frozen; only the MLP head trained) and from-scratch (encoder and MLP head trained jointly with InfoNCE). We report linear-probe accuracy as the primary metric, following SimCLR practice: the encoder is frozen on trained features (5k images), evaluated on test of 8k images, and a single linear classifier is trained on top to isolate representation quality independent of head capacity. We also include $k$-NN classification with $k{=}200$ from a train-feature bank. Table 1 reports mean±std over three seeds: pretrained–frozen attains $77.9 \pm 0.8\%$ (linear probe) and $78.7 \pm 0.8\%$ ($k$-NN), while from-scratch attains $39.7 \pm 0.3\%$ (linear probe) and $53.1 \pm 0.6\%$ ($k$-NN).

| Regime | Linear Probe Acc. (%) | k-NN Acc. (%) |
|---|:---:|:---:|
| Pretrained-frozen | $77.9 \pm 0.8$ | $78.7 \pm 0.8$ |
| From-scratch | $39.7 \pm 0.3$ | $53.1 \pm 0.6$ |

Table 1: Classical baseline results on STL-10. Reported as mean $\pm$ standard deviation over three seeds.

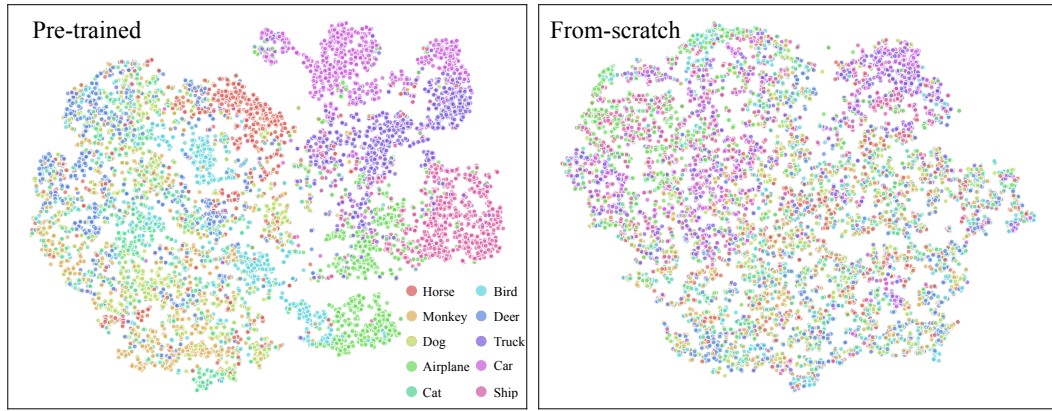

Figure 1: t-SNE visualization of embeddings produced by the classical baseline MLP head. Left: pretrained-frozen encoder. Right: from-scratch encoder. Clearer clustering is observed in the pretrained regime.

Figure 1 compares the embedding space learned by the classical MLP head under two training regimes. In the pretrained-frozen setting (left), the t-SNE projection (van der Maaten & Hinton, 2008) reveals well-defined clusters corresponding to distinct object categories, with minimal overlap between classes such as airplane, bird, and deer. This indicates that the pretrained encoder provides a strong feature basis from which the MLP can extract class-discriminative representations. In contrast, when trained entirely from scratch (right), the embedding space lacks clear separation: clusters are diffuse, class boundaries are poorly defined, and several categories overlap substantially. These results highlight the value of pretraining, as frozen encoders not only accelerate downstream learning but also yield embeddings that are more linearly separable, directly supporting the stronger quantitative results reported in Table 1.

## 3.2 EXPERIMENT 1: VARIATIONAL QUANTUM CIRCUIT (VQC) PROJECTION HEAD

We evaluate the VQC head on STL-10 under the same training protocol as the classical baseline. To avoid confounding from pretrained features, we report results in the *from-scratch* regime (Table 2); preliminary pretrained–frozen runs showed a "dead-wire" behavior in which the lift layer (linear expansion before the quantum circuit) absorbed most of the learning signal, rendering the VQC effectively idle (see Discussion). Hyperparameters (augmentations, optimizer, temperature, batch size, epochs) are held fixed across heads. With a Z-only readout at 8 qubits and one layer, VQC achieves $27.6\pm1.67\%$ LP and $49.13\pm0.54\%$ $k$-NN at $\sim 200$ s/batch; switching to a multi-observable X/Y/Z readout at 8 qubits, one layer improves to $33.57 \pm 0.59\%$ LP and $53.37 \pm 0.13\%$ $k$-NN at $\sim 245$ s/batch. Increasing width to 12 qubits (X/Y/Z, one layer) yields $34.25 \pm 1.31\%$ LP and $52.65 \pm 0.30\%$ $k$-NN with $\sim 460$ s/batch, while increasing depth to two layers at 8 qubits gives $32.45 \pm 0.43\%$ LP and $52.79 \pm 0.19\%$ $k$-NN at $\sim 300$ s/batch. The classical MLP baseline attains $39.7 \pm 0.3\%$ LP and $53.1 \pm 0.6\%$ $k$-NN at $\sim 30$ s/batch, and a capacity-matched bottleneck linear control ($512 \rightarrow n_q \rightarrow 128$) reaches $37.07 \pm 0.65\%$ LP and $52.34 \pm 0.21\%$ $k$-NN with similar latency to the MLP.

| Head | Linear Probe Acc. (%) | $k$-NN Acc. (%) | Latency (s/batch) |
|---|---|---|---|
| MLP (baseline) | $39.7 \pm 0.3$ | $53.1 \pm 0.6$ | $\approx 30$ |
| VQC (Z–only; 8q, $1n_l$) | $27.6 \pm 1.67$ | $49.13 \pm 0.54$ | $\approx 200$ |
| VQC (X/Y/Z; 8q, $1n_l$) | $33.57 \pm 0.59$ | $53.37 \pm 0.13$ | $\approx 245$ |
| VQC (X/Y/Z; 12q, $1n_l$) | $34.25 \pm 1.31$ | $52.65 \pm 0.3$ | $\approx 460$ |
| VQC (X/Y/Z; 8q, $2n_l$) | $32.45 \pm 0.43$ | $52.79 \pm 0.19$ | $\approx 300$ |
| Bottleneck Linear ($n_q \rightarrow 128$) | $37.07 \pm 0.65$ | $52.34 \pm 0.21$ | $\approx 30$ |

Table 2: Comparison of the classical MLP head and VQC heads on STL-10 in From-scratch regime. Mean $\pm$ standard deviation over three seeds. The multi–observable readout expands the quantum head from $n_q$ to $3n_q$ outputs and improves downstream accuracy, while increasing latency with circuit width/depth.

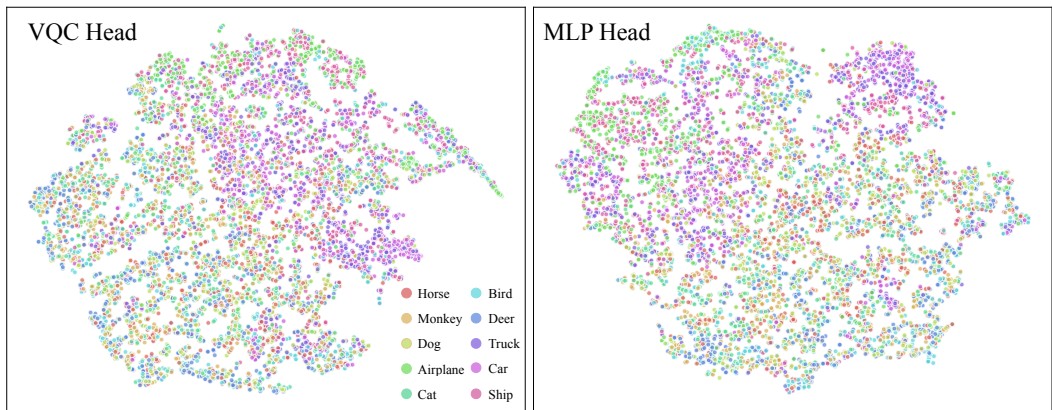

Figure 2: t-SNE embeddings for the VQC head with multi–observable readout (left) and the classical MLP head (right) under matched training protocol. The multi–observable VQC reduces the dead–wire effect observed with Z–only readout and yields clearer class structure, though overall quality remains sensitive to circuit width and depth.

Figure 2 contrasts the embedding quality of the VQC head with multi–observable readout against the classical MLP head under an identical training setup. The VQC embeddings (left) exhibit more coherent class clusters, suggesting that multi–observable readout mitigates the dead–wire problem seen in Z–only measurements and allows information from the encoder to propagate more effectively. Although separation remains imperfect, categories such as airplane and truck are more distinctly isolated compared to the MLP baseline (right), where clusters are fuzzier and overlap is more pronounced. These improvements demonstrate that multi–observable quantum readouts can enhance representational structure, albeit with sensitivity to circuit hyperparameters such as width and depth.

Lastly, we quantify the similarity between representations learned by the classical MLP head and the quantum VQC head using *Centered Kernel Alignment* (CKA) (Kornblith et al., 2019). We report both *linear CKA* (computed directly from feature matrices) and *kernel CKA* (computed from cosine Gram matrices), under two training regimes: (i) a pretrained–frozen encoder and (ii) a from-scratch encoder trained jointly with the head. As shown in Table 3, CKA values are consistently low ($\approx 0.2$) across regimes, indicating limited alignment between the two heads' embeddings despite identical inputs from the backbone.

| Training Regime | Linear CKA | Kernel CKA (cosine) |
|---|---|---|
| Pretrained-frozen encoder | $0.228 \pm 0.094$ | $0.22 \pm 0.103$ |
| From-scratch encoder | $0.279 \pm 0.089$ | $0.238 \pm 0.095$ |

Table 3: CKA similarity between MLP and VQC projection-head representations reported as Mean $\pm$ standard deviation over three random seeds. Linear CKA is computed on feature matrices; Kernel CKA is computed on cosine similarity Gram matrices.

### 3.3 EXPERIMENT 2: QUANTUM FEATURE MAPS (QFM)

We use the same encoder, data pipeline, and optimizer as the baseline; the projection head is removed and InfoNCE operates on fidelity similarities $K_{ij}$ from the QFM in Sec. 2.3. We evaluate pretrained–frozen and from-scratch encoders, sweeping $n_q \in \{4, 8\}$ and $L \in \{2, 3\}$ at fixed temperature $\tau = 0.5$ with angles $\tanh(\cdot)\pi$. Per run we log per-epoch loss, the off-diagonal mean and standard deviation of $K$, and the positive–negative gap $\Delta$ under the NT-Xent layout; after training we report STL-10 linear-probe and $k$-NN ($k=200$) accuracy. Table 4 summarizes mean$\pm$std over three seeds (kernel statistics averaged over the last five epochs). In the pretrained regime, the highest scores occur at $n_q=4$, $L=2$ with raw fidelities (LP $72.50\pm0.01\%$, $k$-NN $75.67\pm0.04\%$); FS–RBF at the same setting yields nearly identical metrics. Increasing $n_q$ and/or $L$ lowers LP and $k$-NN (e.g., $n_q=8$, $L=2$ gives LP $60.42\pm0.03\%$). From-scratch models follow the same pattern with lower absolute accuracies (e.g., LP $\sim 37\%$ at $n_q=4$, $L=2$). Across settings, InfoNCE exceeds MMD with the same kernels. Off-diagonal means of $K$ lie in the range 0.08–0.19 with small standard deviations, and $\Delta$ is consistently high for InfoNCE runs ($\approx 0.65$–0.74).

| Regime | Setting | Kernel | Loss | $\tau / \sigma^2$ | LP (%) | k-NN (%) | off-diag $\mu \pm \sigma$ | $\Delta$ |
|---|---|---|---|---|---|---|---|---|
| Pre. | $n_q=4$, $L=2$ | Fidelity | InfoNCE | $\tau=0.5$ | $72.5 \pm 0.01$ | $75.67 \pm 0.04$ | $0.1 \pm 0.001$ | $0.74 \pm 0.02$ |
| Pre. | $n_q=4$, $L=2$ | FS–RBF | InfoNCE | $\sigma^2=2.09 \pm 0.02$ | $72.46 \pm 0.01$ | $75.67 \pm 0.04$ | $0.1 \pm 0.001$ | $0.74 \pm 0.02$ |
| Pre. | $n_q=4$, $L=2$ | Fidelity | MMD | — | $53.73 \pm 0.03$ | $61.14 \pm 0.03$ | $0.16 \pm 0.02$ | $0.38 \pm 0.02$ |
| Pre. | $n_q=8$, $L=2$ | Fidelity | InfoNCE | $\tau=0.5$ | $60.42 \pm 0.03$ | $68.71 \pm 2.4$ | $0.08 \pm 0.003$ | $0.71 \pm 0.007$ |
| Pre. | $n_q=8$, $L=3$ | FS–RBF | InfoNCE | $\sigma^2=2.29 \pm 0.004$ | $56.04 \pm 0.02$ | $66.27 \pm 0.02$ | $0.08 \pm 0.007$ | $0.7 \pm 0.008$ |
| Pre. | $n_q=8$, $L=3$ | Fidelity | MMD | — | $40.38 \pm 0.03$ | $55 \pm 0.02$ | $0.13 \pm 0.03$ | $0.35 \pm 0.017$ |
| Scr. | $n_q=4$, $L=2$ | Fidelity | InfoNCE | $\tau=0.5$ | $37.4 \pm 0.003$ | $53.9 \pm 0.01$ | $0.11 \pm 0.02$ | $0.69 \pm 0.003$ |
| Scr. | $n_q=4$, $L=2$ | FS–RBF | InfoNCE | $\sigma^2=2.02 \pm 0.02$ | $37.36 \pm 0.003$ | $53.85 \pm 0.005$ | $0.11 \pm 0.002$ | $0.69 \pm 0.003$ |
| Scr. | $n_q=4$, $L=2$ | Fidelity | MMD | — | $34.41 \pm 0.005$ | $52.54 \pm 0.005$ | $.19 \pm 0.009$ | $0.55 \pm 0.007$ |
| Scr. | $n_q=8$, $L=2$ | Fidelity | InfoNCE | $\tau=0.5$ | $37.5 \pm 0.01$ | $54 \pm 0.01$ | $0.09 \pm 0.02$ | $0.65 \pm 0.054$ |
| Scr. | $n_q=8$, $L=3$ | FS–RBF | InfoNCE | $\sigma^2=2.21 \pm 0.04$ | $35.86 \pm 0.006$ | $54 \pm 0.01$ | $0.08 \pm 0.003$ | $0.69 \pm 0.004$ |
| Scr. | $n_q=8$, $L=3$ | Fidelity | MMD | — | $33.13 \pm 0.01$ | $53.01 \pm 0.008$ | $0.18 \pm 0.004$ | $0.55 \pm 0.01$ |

Table 4: Quantum Feature Map (QFM) results on STL-10. Mean $\pm$ std over three seeds. Off-diagonal $\mu \pm \sigma$ is computed on the kernel matrix excluding the diagonal; $\Delta$ is the mean positive–negative gap.

## 4 DISCUSSION

### 4.1 EXPERIMENT 0: CLASSICAL BASELINE

The pretrained–frozen regime exhibits strong linear separability on STL-10, consistent with prior SimCLR-style results, whereas performance drops substantially when training from scratch on this small dataset. This gap clarifies how to interpret our quantum experiments: in the pretrained–frozen setting, downstream heads predominantly refine already informative features (risking "dead-wire" behavior for weakly expressive heads), while in the from-scratch setting the head must actively shape representational geometry under limited data and stricter optimization, providing a more sensitive testbed for potential quantum gains. Accordingly, we use the pretrained–frozen scores as a ceiling reference and focus comparative analysis on from-scratch runs in subsequent sections.

### 4.2 EXPERIMENT 1: VARIATIONAL QUANTUM CIRCUIT (VQC) PROJECTION HEAD

Table 2 indicates three main effects. First, measurement expressivity matters: moving from a Z-only to an X/Y/Z readout (expanding the head from $n_q$ to $3n_q$ outputs) consistently narrows the

gap to the MLP, suggesting that readout richness, not just circuit depth, is a primary bottleneck. Second, scaling width/depth increases latency sharply (from $\sim 200$ to $\sim 460$ s/batch across the shown grid) without delivering commensurate accuracy gains, reinforcing that current simulators constrain practical VQC operating regimes. Third, the bottleneck linear control recovers much of the MLP's performance at MLP-like cost, implying that part of the observed uplift comes from projection capacity rather than uniquely quantum transformations. We therefore focus on from-scratch results; in the pretrained–frozen setting the lift layer preceding the VQC absorbed the available gradient signal and the circuit behaved as a "dead wire," masking any incremental benefit from the quantum module. Overall, while multi-observable VQCs show meaningful dependence on qubit count and improve over Z-only designs, the MLP head remains stronger under our budgets, with VQC latency scaling as the dominant practical limitation.

A practical takeaway from our latency measurements shown in Table 2 is a simple scaling rule of thumb for when VQC heads are worth revisiting on simulators. With parameter–shift and statevector backends, per–batch time grows roughly like $B \, 2^{n_q} (n_q L)^2$ (constants suppressed); anchored to our measurement of $\sim 200$ s/batch at $n_q{=}8$, $L{=}1$, this suggests a feasible envelope near $n_q \leq 6$, $L \leq 1$ (approximately keeping $n_q L \lesssim 8$) if one aims to stay within tens of seconds per batch. Faster differentiation (e.g., adjoint) and true circuit batching would relax the scaling toward $B_{\text{eff}} \, 2^{n_q} (n_q L)$, materially enlarging the search space; absent these improvements, deeper/wider circuits quickly become time–prohibitive regardless of accuracy. As a coarse guideline, VQCs become compelling to revisit when simulator and/or batching upgrades yield a net $\geq 4$–$8\times$ speedup over our anchor, or when modest low–latency hardware can keep batch times in the 30–60 s range. *Side note:* these thresholds are hardware–dependent; see Appendix C for the laptop configuration used in our experiments.

Lastly, Table 3 shows CKA values near zero imply that two representations are largely dissimilar, whereas values near one indicate strong alignment up to invertible linear transforms and isotropic rescaling. The persistently low CKA observed here suggests that the VQC head induces an embedding geometry that is substantially different from the MLP head. In the pretrained-frozen case, this points to a projection mismatch given identical encoder inputs; in the from-scratch case, it implies that joint optimization with a VQC head biases the learned space away from the MLP solution manifold. Together with downstream metrics, these results support the view that the present VQC configuration either (a) lacks capacity to preserve or shape the encoder's structure in a manner similar to the MLP, or (b) optimizes toward a qualitatively different (and possibly less informative) geometry. This motivates exploring more expressive variational ansätze, richer measurement sets, or hybrid heads that better couple to the encoder's feature statistics.

### 4.3 EXPERIMENT 2: QUANTUM FEATURE MAPS (QFM)

Table 4 indicates that, under analytic simulation and standard budgets, QFM–InfoNCE tracks but does not surpass the classical cosine baseline. The decline in accuracy with larger $n_q/L$ aligns with kernel concentration signals (low off-diagonal means with narrow spread) (Huang et al., 2021), suggesting reduced discriminability as depth/qubits increase. High $\Delta$ despite small raw fidelities implies that NT-Xent can separate positives from negatives even when the kernel's dynamic range is limited, but the resulting representation quality plateaus. FS–RBF yields outcomes similar to raw fidelities and is sensitive to bandwidth selection, while MMD underperforms InfoNCE with the same kernels, indicating that contrastive normalization and temperature scaling are beneficial in this regime. Overall, these results support a conservative reading: the particular angle-encoding, shallow ring-entangled maps explored here do not provide a measurable uplift on STL-10; improving quantum-induced similarity likely requires better-conditioned feature maps (e.g., alternative encodings or data re-uploading schemes that avoid similarity collapse), stronger pre-normalization of encoder features, or different temperatures, alongside engineering advances that permit broader sweeps (larger grids, longer schedules) without prohibitive latency.

## 5 CONCLUSION

We presented a systematic empirical study of two common quantum insertions into a SimCLR-style contrastive pipeline on STL-10: (i) a variational quantum circuit (VQC) used as the projection head and (ii) a fixed quantum feature map (QFM) whose state fidelities replace cosine similarity

in InfoNCE. Experiments were run in realistic simulator regimes (few qubits, shallow depth) with classical baselines matched for data and optimization.

After observing in the pretrained–frozen setting that a preceding lift layer absorbed most of the learning signal, leaving the VQC effectively a "dead wire", we used from-scratch training as the primary testbed. In this setting, the VQC head did not outperform a tuned MLP head; increasing qubits or depth sharply increased latency without commensurate accuracy gains, reflecting statevector throughput limits and parameter-shift costs. For QFMs, fidelity-based similarity was a stable drop-in but closely tracked cosine similarity, yielding no measurable uplift consistent with our kernel-concentration diagnostics.

These negative results are informative: they narrow the plausible claim space for near-term quantum components in contrastive learning and indicate where progress would matter most. Concretely, higher-throughput, batching-friendly/adjoint simulators; lower-variance gradient estimators and better-conditioned encodings that avoid similarity collapse; and modest, low-latency hardware backends could open training regimes that current software cannot emulate efficiently.

Finally, to address external validity beyond STL-10, we will include in the supplementary material a minimal CIFAR-10/100 replication that mirrors our protocol (same encoder and linear-probe evaluation, reduced hyperparameter sweeps). The goal is not exhaustive tuning, but to test whether the VQC latency–accuracy trade-offs and the fidelity-versus-cosine behavior seen on STL-10 also appear on these standard benchmarks.

## 6 REPRODUCIBILITY STATEMENT

We provide the materials necessary to reproduce our results. Model architectures (ResNet18 encoder, MLP head, and VQC head), quantum feature maps/kernels, and training procedures are described in Sect. 2. Implementation details and hyperparameters (optimizer settings, learning rates, batch sizes, temperature, augmentation pipeline, qubit counts/layers, entanglement patterns) are consolidated in Appendix A. Dataset sources and preprocessing steps for STL-10 and any additional benchmarks are documented in Appendix B. An anonymized code archive is included in the supplemental materials with training/evaluation scripts, configuration files with fixed random seeds, and environment specifications (version pins for PyTorch (Paszke et al., 2019), PennyLane (Bergholm et al., 2022), and Qiskit (Aleksandrowicz et al., 2019)), along with shell scripts to regenerate all tables and figures. Instructions for verifying numerical stability (e.g., precision settings) and for reproducing figures from logged artifacts are provided in the supplemental `Reproducibility.md` file, which also contains the link to the anonymized repo.

## 7 ETHICS STATEMENT

This work uses publicly available datasets and simulators, and it does not involve human subjects or personal data.

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

# A  HYPERPARAMETERS

This appendix consolidates all training and evaluation hyperparameters referenced in the main text. We group them into (i) common settings, (ii) classical (MLP head) settings, and (iii) quantum (VQC head and quantum kernels/feature maps) settings. Tables 5–7 list the exact fields you need to reproduce each experiment; Section numbers in the rightmost column indicate where each item is discussed in the paper. When a value was swept, we list the grid; the default used for the main report is **bolded**. Random seeds are fixed unless otherwise noted.

**Environment and versions.**  We pin library versions used in all reported runs: PyTorch, Torchvision, PennyLane, and Qiskit, with exact version strings and CUDA/toolchain details recorded in the project's `requirements.txt` (or `environment.yml`) included in the supplemental code archive. Reproduction scripts (`run.sh`, `make_figures.sh`) invoke the configurations behind Tables 5–7 with fixed random seeds.

**Determinism settings.**  Unless otherwise stated, we set `torch.backends.cudnn.benchmark=False` and enable deterministic algorithms where available; any unavoidable nondeterminism (e.g., from low-level kernels) is documented in the code README.

| Category | Hyperparameter | Value(s) | Notes |
|---|---|---|---|
| Dataset | Benchmark | STL-10 | Unlabeled for pretraining; standard labeled split for linear eval. |
| Dataset | Resolution / Norm. | $96{\times}96$; mean/std $(0.5, 0.5, 0.5)$ | As in common STL-10 setups. |
| Augmentation | Pipeline | RRC, HFlip, CJitter, Gray, Blur | See Table 6 for probabilities/ranges. |
| Optimization | Optimizer | AdamW ($\beta{=}(0.9, 0.999)$) | |
| Optimization | Learning rate | **3e-4** | Cosine decay; warmup = 10% of steps. |
| Optimization | Weight decay | **1e-4** | Excludes bias/norm params. |
| Optimization | Batch size | **256** | Global (use accumulation if needed). |
| Training | Epochs | **200** | |
| Training | Grad clip | None | If enabled: norm = 1.0. |
| Contrastive | Temperature $\tau$ | **0.1** | InfoNCE temperature. |
| Precision | Mixed precision | **fp16** | PyTorch AMP with loss scaling. |
| Determinism | Seeds | **42** | Torch/NumPy/PL seeds; `cudnn.benchmark=False`. |
| Eval | Linear probe | Logistic regression / 100-epoch linear head | Frozen backbone; early stopping on val acc. |

Table 5: Common hyperparameters (data, optimization, evaluation).

| Transform | Param | Value(s) | Notes |
|---|---|---|---|
| RandomResizedCrop | scale; ratio | **[0.2,1.0]**; [3/4,4/3] | |
| HorizontalFlip | prob | **0.5** | |
| ColorJitter | $(b, c, s, h)$; prob | **(0.4,0.4,0.4,0.1)**; **0.8** | Before grayscale. |
| Grayscale | prob | **0.2** | |
| GaussianBlur | kernel; $\sigma$; prob | 23; **[0.1,2.0]**; **0.5** | Kernel may scale with image size. |
| Normalization | mean/std | **(0.5,0.5,0.5) / (0.5,0.5,0.5)** | After tensor conversion. |

Table 6: Augmentation hyperparameters.

| Component | Hyperparameter | Value(s) | Notes |
|---|---|---|---|
| Backbone | Encoder | ResNet18 | Torchvision baseline; use pooled features. |
| Backbone | Feature dim $d$ | 512 | ResNet18 output width. |
| MLP head | Hidden dim | **2048** | Two-layer MLP unless stated. |
| MLP head | Output dim | **128** | Normalized embeddings. |
| MLP head | Nonlinearity / Norm | ReLU; BN/LN | Applied between/after linears as specified. |
| Bottleneck (alt.) | Linear($n_q \rightarrow 128$) | optional | Used for comparisons with VQC head. |
| VQC head | Qubits $n_q$ | **4** ($\{2,4,6,8\}$) | Circuit wires. |
| VQC head | Layers $L$ | **2** ($\{1,2,3\}$) | Rotation + entanglement blocks. |
| VQC head | Rotations | RX/RY/RZ | Angle embedding from linear map of $h \in \mathbb{R}^d$. |
| VQC head | Entanglement | Ring | CZ/CNOT ring per layer. |
| VQC head | Measurement | $\langle Z \rangle$ | Aggregated to $\mathbb{R}^{n_q}$ then normalized. |
| VQC head | Shots | **None** | Analytic; if stochastic, 1024 shots. |
| VQC head | Output dim | $n_q$ or **128** | Optional classical bottleneck. |
| VQC head | Backend | PL default.qubit / Qiskit Aer | Exact simulator unless noted. |
| Quantum kernels | Feature map depth | **2** | Angle-encoding depth for kernel experiments. |
| Quantum kernels | Similarity | Fidelity / inner product | Used inside InfoNCE. |
| Stability | Gradients | Parameter-shift | Default in PennyLane for diff. circuits. |

Table 7: Encoder + heads: classical MLP and quantum VQC settings.

## B  DATASETS AND PREPROCESSING

This appendix documents dataset sources and the exact preprocessing pipelines used in pretraining and evaluation. When applicable, we cite canonical hosting locations and provide the transforms in functional form so that the same behavior can be reproduced with common libraries (e.g., `torchvision`). No custom curation or filtering was applied beyond what is stated here.

**STL-10 (source, protocol, and preprocessing).** STL-10 is a 10-class natural image dataset with $96 \times 96$ RGB images comprising an unlabeled split (100,000 images) and a labeled train/test split (5,000/8,000 images); we use the official release via `torchvision.datasets.STL10` (mirrored from the original Stanford CS hosting). For self-supervised pretraining we use only the `unlabeled` split and ignore labels; for linear evaluation we freeze the encoder, fit a linear/logistic head on the official `train` split, and evaluate on the official `test` split; unless noted otherwise, we hold out 10% of the labeled `train` images as validation for temperature tuning and early stopping. Images are decoded with the default PNG/JPEG decoders in `Pillow` and treated as RGB without additional color-space conversions. Pretraining uses two augmented views per image with the pipeline RRC(scale=$[0.2, 1.0]$) $\rightarrow$ HFlip($p$=0.5) $\rightarrow$ ColorJitter($b$=0.4, $c$=0.4, $s$=0.4, $h$=0.1, $p$=0.8) $\rightarrow$ Grayscale($p$=0.2) $\rightarrow$ GaussianBlur($\sigma \in [0.1, 2.0], p$=0.5) $\rightarrow$ ToTensor $\rightarrow$ Normalize($\mu$=(0.5, 0.5, 0.5), $\sigma$=(0.5, 0.5, 0.5)); linear-eval (train) uses RRC(scale=$[0.2, 1.0]$) $\rightarrow$ HFlip(0.5) $\rightarrow$ ToTensor $\rightarrow$ Normalize($\mu, \sigma$); linear-eval (test) uses CenterCrop(96) $\rightarrow$ ToTensor $\rightarrow$ Normalize($\mu, \sigma$). Because STL-10 is natively $96 \times 96$, no resizing is applied beyond random/center crops. Normalization constants are fixed to $(\mu, \sigma)$=(0.5, 0.5, 0.5) to maintain parity across experiments and avoid data-dependent statistics. Data-loader settings are: global batch size 256 (with gradient accumulation if needed), shuffling enabled for training, `pin_memory=True`, and `num_workers` set per machine; random seeds are fixed (Appendix A). No class rebalancing, deduplication, or corruption filtering is applied, and we do not use the historical "folds" protocol; we train on the full labeled train split (minus the validation holdout) for linear probing and report accuracy on the official test split.

## C    COMPUTER HARDWARE

All experiments were executed on a single laptop: Apple M4 Max (Apple Silicon) with 64 GB unified memory running macOS Sequoia 15.6.1. We used PyTorch's Metal Performance Shaders (MPS) backend for the classical encoder when available and CPU execution for quantum circuit simulation (statevector, analytic). Reported wall–clock latencies and throughput reflect this configuration and may vary on other systems (e.g., CUDA GPUs, different CPU cores, or simulators with batched/adjoint differentiation).

## D    LLM DISCLOSURE

We used large language models (LLMs) in three limited ways: (1) to tighten prose and correct grammar; (2) to suggest robustness checks and sanity-check intermediate results and interpretations; and (3) to help diagnose ambiguous errors encountered when using PyTorch, Qiskit, and PennyLane. All outputs from LLMs were reviewed, edited, and validated by the authors, who remain solely responsible for the analyses, code, and conclusions.

