# OpenReview forum: "Contrastive Learning with Quantum Projection Heads and Kernels"
_ICLR.cc/2026/Conference — ICLR 2026 Conference Withdrawn Submission_

### Official Review · Reviewer_uPfa · 2025-10-23

**Soundness:** 1
**Presentation:** 1
**Contribution:** 1
**Rating:** 2
**Confidence:** 5

**Summary:**

This paper presents a systematic empirical study investigating the integration of quantum components into a self-supervised contrastive learning (SimCLR) pipeline. The primary research question is whether replacing classical elements with quantum analogues—specifically, a Variational Quantum Circuit (VQC) with angle encoding as a projection head and a fixed quantum feature map (QFM) for similarity computation—yields a tangible performance benefit.

**Strengths:**

The authors performed experiments with replacing classical components with quantum circuits in the scenario of contrastive learning.

**Weaknesses:**

It provides no contribution to the society and the reasons to cause such empirical results have been well analyzed and understood. The quantum circuits under angle encoding are inherently trucated Fourier series; the difference is that quantum circuits can have exponentially increasing spectrum span. This explains the observations in the experiment. The same goes to the quantum feature map. In summary, everything that is presented in this paper has been well understood. The paper also needs better sturcture layout.

**Questions:**

No.

**Details Of Ethics Concerns:**

No.

---

### Official Review · Reviewer_bgtN · 2025-10-30

**Soundness:** 1
**Presentation:** 2
**Contribution:** 1
**Rating:** 2
**Confidence:** 4

**Summary:**

This paper primarily investigates the effectiveness of quantum projection heads and quantum kernels in contrastive learning.

**Strengths:**

1. The motivation behind this paper is clear.

**Weaknesses:**

1. It is recommended that the authors include a flowchart or framework diagram of the algorithm proposed in the paper.

2. Since the authors' conclusions are based on small-scale quantum systems, it is recommended that they verify them on larger-scale quantum systems (e.g. 20 qubits). Because in small-scale Hilbert spaces, quantum computing may not necessarily have an advantage [1].

3. The paper uses a fixed quantum feature map in experiments, which is inappropriate. And inappropriate quantum feature maps tend to be inferior to that of classical kernels [2,3]. So the reviewer kindly suggest that the authors could use training embedding kernels [4] to conduct all the experiments.

4. To demonstrate the practical performance of quantum machine learning algorithms, the reviewer kindly suggest the authors include experiments with relevant noisy quantum computers and noisy simulators.

5. Reviewer are pleased to see the authors include SVM classifier in the experimental results.

[1] Does provable absence of barren plateaus imply classical simulability?

[2] Power of data in quantum machine learning.

[3] The Inductive Bias of Quantum Kernels.

[4] Training quantum embedding kernels on near-term quantum computers.

**Questions:**

Please see the weaknesses.

---

### Official Review · Reviewer_ywA9 · 2025-10-31

**Soundness:** 4
**Presentation:** 3
**Contribution:** 2
**Rating:** 4
**Confidence:** 4

**Summary:**

This paper evaluates the potential benefits of integrating quantum computing components into a modern self-supervised visual representation learning pipeline (SimCLR-style). The authors replace the classical projection head and similarity measure with quantum counterparts (VQCs and quantum kernels, respectively). The study is carefully controlled and benchmarked against classical MLP heads. The core finding is a "null baseline": under realistic NISQ-era constraints (small circuits), the quantum components provide no performance benefit over a well-tuned classical MLP. The authors also note that scaling quantum circuits quickly becomes computationally prohibitive in simulation. They conclude this baseline is valuable for guiding future research.

**Strengths:**

The study is meticulous in its experimental setup, providing a fair "apples-to-apples" comparison between classical and quantum components. This thoroughness is commendable.
The authors are transparent about their negative results, which adds to the paper's trustworthiness.
Establishing a "null baseline" is indeed valuable for the research community, as it prevents wasted effort and helps set a clear bar for future work to surpass.
The paper correctly identifies practical bottlenecks (e.g., simulation latency) that hinder research in this area.

**Weaknesses:**

In all tested settings, the quantum variants did not outperform a tuned classical MLP head. This is the central finding, and it is a negative one. The paper fails to demonstrate any quantum advantage or even competitive performance, which is a significant weakness for a submission to ICLR.
Only small/shallow quantum circuits were feasible; larger circuits were too slow to simulate. While the authors correctly identify this as a bottleneck, it also means the study was unable to explore the regime where quantum advantage might actually exist. This makes the "null result" conclusion limited to the (already suspected) non-advantageous NISQ regime.
The quantum components explored are fairly standard (a basic VQC head, a fixed quantum kernel). The paper does not propose any new quantum techniques; it only tests existing ones in a new context and finds they do not work.

**Questions:**

1.I suggest the authors state clearly whether any run or variant ever beat the MLP head, even marginally. If not, saying so explicitly is valuable to the community and reinforces the "null" finding.
2.A brief quantitative note on how runtime explodes when increasing qubits/depth would help clarify the current scaling bottleneck.
3.It would help to mention one promising next step (e.g., a trainable quantum feature map rather than a fixed kernel) so readers see a path forward.
4. For completeness, listing the VQC head setup (qubit count, entanglement pattern, optimizer, LR, epochs) in an appendix would address fairness concerns

---

### Official Review · Reviewer_eEBm · 2025-10-31

**Soundness:** 2
**Presentation:** 2
**Contribution:** 2
**Rating:** 2
**Confidence:** 5

**Summary:**

In this paper, the authors study whether plugging quantum components into a standard contrastive/self-supervised pipeline actually helps. Concretely, the authors take a SimCLR-style setup on STL-10 with a ResNet-18 encoder and replace the usual MLP projection head with (i) a variational quantum circuit (VQC) projection head, and (ii) a fixed quantum feature map whose pairwise fidelities are used as the similarity/kernel inside InfoNCE. The authors find several results, including tuned classical MLP heads that match or beat VQC heads; fidelity-based quantum kernels mostly behave like cosine similarity; and as soon as you increase qubits/depth, latency explodes.

**Strengths:**

1. The paper controls for encoder, data pipeline, optimizer, and temperature, so comparisons between classical and quantum heads are fair. The authors also give full hyperparameters and environment details (Appendix A–C), which are suitable for reproducibility.

2. The message “current, realistic quantum components do not help SimCLR on STL-10” is valuable for the community because it narrows the plausible claim space.

**Weaknesses:**

1. The paper shows that (their) VQC heads and (their) quantum kernels don’t beat a classical MLP on STL-10. That is important as a benchmark, but it is also precisely what many people in quantum-ML would expect when the encoder is strong and the quantum budget is tiny. There is no new quantum architecture, no new training rule, no theory on when a quantum head should help, and no scaling law beyond informal latency comments. For ICLR, we usually want at least one of these.

2. The authors conducted everything on STL-10, with ResNet-18, and SimCLR/InfoNCE. The authors mention CIFAR-10/100 in the supplementary, but there is no real multi-dataset narrative in the main paper. This makes it hard to generalize the null result to other modalities or to harder SSL settings.

3. The VQC is angle-encoding + ring entanglement + shallow depth + simple multi-observable readout. That’s reasonable, but the paper then strongly concludes “VQC doesn’t help.” In reality, what they’ve shown is “this VQC, at this depth, with this readout, on this dataset, under analytic simulation, doesn’t help.” That’s weaker than the way the Discussion is written.

4. Table 4 shows fidelities in the 0.08–0.19 off-diagonal range with small variance, and accuracy drops when nq or depth grows. But the paper doesn’t connect this to existing theory to explain that, if you do shallow-angle encoding on already-good features. As a result, the paper reads more like a description than an explanation.

5. All the experiments are classically simulated. Although the authors note that latency is the main bottleneck (200–460 s/batch), the paper still frames it as “a useful near-term baseline."

**Questions:**

1. Can the authors show a second dataset (e.g., CIFAR-100) in the main paper, not only in the supplement, using the same quantum heads, to prove the null result is not STL-10 specific?

2. Can the authors try a data-reuploading VQC, or a hardware-efficient ansatz with mid-circuit measurements, to rule out “you picked an especially weak circuit”?

3. Can the authors show even a tiny hardware run (4 qubits, 1 layer) to validate that the latency story is not simulation-only?

4. Can the authors add a classical “heavy” head (e.g., 2–3× params) to show you are not just capacity-bound?

---

### Note · Authors · 2025-11-17

I have read and agree with the venue's withdrawal policy on behalf of myself and my co-authors.